# Lack of Retinoblastoma Protein Shifts Tumor Metabolism from Glycolysis to OXPHOS and Allows the Use of Alternate Fuels

**DOI:** 10.3390/cells11203182

**Published:** 2022-10-11

**Authors:** Vishnu Suresh Babu, Gagan Dudeja, Deepak SA, Anadi Bisht, Rohit Shetty, Stephane Heymans, Nilanjan Guha, Arkasubhra Ghosh

**Affiliations:** 1GROW Research Laboratory, Narayana Nethralaya Foundation, Bangalore 560099, India; 2Department of Cardiology, Cardiovascular Research Institute Maastricht (CARIM), Maastricht University, 6229 ER Maastricht, The Netherlands; 3Retinoblastoma Service, Narayana Nethralaya, Bangalore 560099, India; 4Agilent Technologies India Pvt Ltd., Bangalore 560048, India; 5Centre for Molecular and Vascular Biology, Department of Cardiovascular Sciences, KU Leuven, Herestraat 49, Bus 911, 3000 Leuven, Belgium

**Keywords:** retinoblastoma, metabolism, glycolysis, OXPHOS, OCR, energetics

## Abstract

Mutations in the *RB1* locus leading to a loss of functional Rb protein cause intraocular tumors, which uniquely affect children worldwide. These tumors demonstrate rapid proliferation, which has recently been shown to be associated with an altered metabolic signature. We found that retinoblastoma tumors and in-vitro models lack Hexokinase 1 (HK1) and exhibit elevated fatty acid oxidation. We show that ectopic expression of *RB1* induces HK1 protein in Rb null cells, and both *RB1* and *HK1* can mediate a metabolic switch from OXPHOS to glycolysis with increased pyruvate levels, reduced ATP production and reduced mitochondrial mass. Further, cells lacking Rb or HK1 can flexibly utilize glutamine and fatty acids to enhance oxidative phosphorylation-dependent ATP generation, as revealed by metabolic and biochemical assays. Thus, loss of Rb and HK1 in retinoblastoma reprograms tumor metabolic circuits to enhance the glucose-independent TCA (tricarboxylic acid) cycle and the intermediate NAD+/NADH ratios, with a subsequent increase in fatty-acid derived L-carnitine to enhance mitochondrial OXPHOS for ATP production instead of glycolysis dependence. We also demonstrate that modulation of the Rb-regulated transcription factor E2F2 does not result in any of these metabolic perturbations. In conclusion, we demonstrate *RB1* or *HK1* as critical regulators of the cellular bioenergetic profile and identify the altered tumor metabolism as a potential therapeutic target for cancers lacking functional Rb protein.

## 1. Introduction

Retinoblastoma (Rb) is the most common pediatric ocular malignancy which is caused by mutations in the *RB1* gene [1]. Rapidly growing Rb tumors are fed by feeder arteries and drainage veins [2] to facilitate choroidal, scleral and optic nerve invasion [3,4]. Rb tumors possess an aggressive phenotype that is driven not only by the lack of cell cycle checkpoints but also by metabolic alterations [5,6], a consequence of functional *RB1* loss [7]. Reprogramming of metabolic circuits in tumors favors their growth, and such changes in individual metabolic pathways frequently correlate with enhanced biosynthesis of energy substrates [8]. The lack of Rb protein is known to enhance mitochondrial function [9] and alter metabolic pathways in various cancers [10,11] to meet tumor energy requirements [12].

Mitochondrial oxidative phosphorylation (OXPHOS) generates cellular energy (ATP) by utilizing reduced intermediates derived from glucose, glutamine, or fatty acid substrates for efficient energy production [13]. Glucose and glucose-derived pyruvate are the favorite fuel choices for various cancers [14], while under restrictive nutrient conditions, some tumors alter their metabolic circuits to use glutamine and fatty acids to satisfy their bioenergetic demands [15,16]. However, in the context of retinoblastoma, the mechanisms of energy production and the fuel choices in the tumor cells are not well defined. Previous reports and our recent discovery demonstrate that Rb tumors are associated with high fatty-acid metabolism and low glycolysis [17,18]. We have identified a set of altered metabolic pathways in Rb patient tumors and *RB1* null in vitro models [17]. Further, Rb protein depletion enhances glutamine uptake in cells [19], but the metabolic flexibility of Rb tumors to utilize reduced intermediates derived from various metabolic pathways remains unexplored. Therefore, in this study, using metabolic assays, we have unraveled the metabolic phenotype of Rb null tumor cells and their usage of alternate fuels to meet energy requirements. We hypothesized that the lack of *RB1* in Rb tumor cells mediates enhancement of glycolysis independent energy production, primarily via enhanced mitochondrial function; therefore, Rb tumor cells possibly switch to alternate fuels and energy production mechanisms such as mitochondrial fat oxidation [20]. Therefore, we studied in detail, using a variety of metabolic and biochemical assays, the cellular bioenergetics in two Rb null cell lines along with a primary, patient-derived Rb null cell line.

## 2. Materials and Methods

### 2.1. Clinical Samples

The study was conducted by the Declaration of Helsinki principles under a protocol approved by the institutional ethics committee of Narayana Nethralaya (EC Ref no: C/2013/03/02). Informed written consent was received from the patient’s guardians before inclusion in the study. The GL1-RB1 line was developed from an enucleated tumor specimen obtained from the right eye of a 3-year-old female retinoblastoma subject (AJCC staging-cT4b). The tumor tissue was enzymatically dissociated using dispase and trypsin, and the cells were cultured in RPMI 1640 media (Cat #11875093, Gibco, Grand Island, NY, USA) supplemented with 10% FBS (cat#A4766801, Gibco, Grand Island, NY, USA), 1% pen strep (cat#15070063, Gibco, Grand Island, NY, USA) and a 10 ng cocktail of EGF (cat#01-407, Sigma Aldrich, St. Louis, MO, USA), VEGF (cat#01-185, Sigma Aldrich, St. Louis, MO, USA) and FGF (cat#GF003AF, Sigma Aldrich, St. Louis, MO, USA). A biphasic population of primary Rb cells was observed during the first two weeks, comprising retinoblastoma tumor spheres adherent to feeder fibroblasts and suspension clusters of single-cell retinoblastoma. Over four weeks, the tumor spheres detached from the fibroblast and formed an unusual chain of suspension cells. These cells were cultured separately, and cell population doubling time was calculated from the exponential growth phase curve. The details of FFPE clinical samples used for mitochondrial staining, including age, gender, laterality, tumor viability, and clinical and histopathology details, are mentioned in Table 1.

### 2.2. Cell Lines

WERI-Rb1 and Y79 cells were obtained from the American Type Culture Collection (ATCC, Manassas, VA, USA). The WERI-Rb1 and Y79 cells were cultured in RPMI 1640 medium (Gibco, Grand Island, NY, USA, Cat #11875093) supplemented with 10% FBS (cat# A4766801, Gibco, Grand Island, NY, USA) and 1% Pen Strep (penicillin–streptomycin) (cat#15070063, Gibco, Grand Island, NY, USA) and maintained at 37 °C in a humidified atmosphere of 5% CO_2_, with intermittent shaking in an upright T25 flask.

### 2.3. Gene Expression Analysis

Total RNA was isolated from cells using the Trizol reagent (Invitrogen, Carlsbad, CA, USA) according to the manufacturer’s protocol. A total of 1µg of RNA was reverse transcribed using Bio-Rad iScript cDNA synthesis kit (cat#1708890, Bio-Rad, Hercules, CA, USA), and quantitative real-time PCR was performed using Kappa Sybr Fast qPCR kit (cat#KK4601, Kapa Biosystems Pty (Ltd.), Cape Town, South Africa) using a Bio-Rad CFX96 system. Relative mRNA expression levels were quantified using the ∆∆C(t) method. Results were normalized to housekeeping human β-actin. Details of primers used are described in Table 2.

### 2.4. Lentiviral Plasmids and Vectors

We constructed a lentiviral plasmid expressing the *RB1* gene in the pCL20 backbone [17]. We purchased commercially available overexpression plasmids for *E2F2* (cat#TOLH-1508827, Transomics Technologies Inc., Huntsville, AL, USA) and *HK1* (cat#TOLH-1505162, Technologies Inc, Huntsville, AL, USA) in a pLX304 lentiviral backbone having CMV promoter. The corresponding shRNA constructs for *E2F2* and *HK1* were in the pZIP lentiviral backbone containing CMV promoter (Transomics Technologies Inc, Huntsville, AL, USA), and the target sequences are available in Appendix A. Lentiviral transduction was used for *RB1, E2F2, HK1* overexpression and knockdown in cell lines using the previously described protocol [21]. Lentivirus was produced in HEK 293T cells, and the media supernatant was concentrated by centrifugation. A total of 1 × 10^6^ WERI-Rb1 cells in 1 mL serum-free media were transduced with 50 µL of 100× *g* concentrated lentiviral preparations of *RB1, E2F2, HK1* and their knockdown viruses in 6 well plates for 4 h, with intermittent shaking at every 30 min. The specific gene expression efficiencies were determined using RT-PCR after 72 h.

### 2.5. Western Blotting

For Western blot analysis, cells were lysed in RIPA buffer (20 mM Tris pH 8.0, 0.1% SDS, 150 mM NaCl, 0.08% sodium deoxycholate, and 1% NP40) supplemented with 1 tablet of protease inhibitor (complete ultra mini-tablet, Roche, Indianapolis, IN, USA) and phosphatase inhibitor (PhosStop tablet, Roche, Indianapolis, IN, USA). A total of 20 µg of total protein was loaded per lane and separated by SDS-PAGE. The separated proteins on the gel were transferred onto the PVDF membrane and were probed for specific antibodies against Rb (cat#9309; Cell Signaling Technology, Danvers, MA, USA), phospho-Rb (cat#8516, Cell Signaling Technology, Danvers, MA, USA), E2F2 (ab209662; Abcam, Cambridge, UK), HK1(cat#2024; Cell Signaling Technology, Danvers, MA, USA) and GAPDH (cat#5174; Cell Signaling Technology, Danvers, MA, USA) at 1:1000 dilution in 5%BSA in 1× TBST overnight at 4 °C. After 4 washes with 1× TBST for 10 min, membranes were incubated with HRP-conjugated anti-mouse (cat#7076; Cell Signaling Technology, Danvers, MA, USA) or anti-rabbit antibodies (cat#7074; Cell Signaling Technology, Danvers, MA, USA) at 1:2000 dilution for 2 h. Images were visualized using the Image Quant LAS 500 system (GE Healthcare Life Sciences, Piscataway, NJ, USA).

### 2.6. Energy Phenotype Assay

Cells were seeded onto an XFp 8-well Seahorse XFp flux plate (cat#103723-100, Agilent Technologies, Santa Clara, CA, USA) pre-coated with 0.01% poly-L-lysine (cat#P4707, Sigma Aldrich, St. Louis, MO, USA). A cell density of 4000 cells/150 µL per well was seeded and centrifuged at 500× *g* rpm to encourage adhesion to the plate and form an evenly dispersed monolayer. Cells were then incubated at 37 °C in non-CO_2_ conditions in Seahorse XF assay medium (cat#103681-100, Agilent Technologies, Santa Clara, CA, USA) and further processed using the XFp Extracellular Flux Analyzer (cat#431018, Agilent Technologies, Santa Clara, CA, USA) as per the manufacturer’s protocols. The energetic profile of modulated and control cells was determined using Seahorse XFp Extracellular Flux Analyzer and Cell Energy Phenotype Test Kit (cat#103275-100, Agilent Technologies, Santa Clara, CA, USA).

### 2.7. Mitochondrial Stress Assay

Cells were seeded onto an XFp 8-well Seahorse XFp flux plate (cat#103723-100, Agilent Technologies, Santa Clara, CA, USA) pre-coated with 0.01% poly-L-lysine. A cell density of 4000 cells/150 µL per well was seeded and centrifuged at 500× *g* rpm to encourage adhesion to the plate and form an evenly dispersed monolayer. Cells were then incubated at 37 °C non-CO_2_ conditions in Seahorse XF assay medium (cat#103681-100, Agilent Technologies, Santa Clara, CA, USA) and further processed using the XFp Extracellular Flux Analyzer (cat#431018, Agilent Technologies, Santa Clara, CA, USA) as per the manufacturer’s protocols. Mitochondrial function was measured using a Seahorse XF Mito-stress test kit (cat#103010-100, Agilent Technologies, Santa Clara, CA, USA). Briefly, mitochondrial OCR (oxygen consumption rate) was measured after injections of 0.5 μM oligomycin, 1 μM FCCP, 1 μM antimycin A, and 1 μM rotenone, according to the manufacturer’s instructions. Determinants of mitochondrial function (basal respiration, maximal respiration, spare respiratory, and ATP production) were calculated using the formulas according to manufactures protocol. All measurements were normalized to a total number of cells using the PrestoBlue Cell Viability Reagent (cat#A13261, Invitrogen, Waltham, MA, USA) post-mitochondrial stress assay. Data were analyzed using Seahorse XFp Wave Software (Version 2.4, Agilent Technologies, Santa Clara, CA, USA) and expressed as replicate data points ± SD of triplicate experiments.

### 2.8. Glycolytic Rate Assay

For glycolytic rate analysis in WERI-Rb1 cells of different conditions, the cells were seeded onto a 0.01% poly-L-lysine-coated XFp 8-well flux plate (cat#103723-100, Agilent Technologies, Santa Clara, CA, USA) in Seahorse XF glycolytic assay conditioned medium (Seahorse XF assay medium cat#103576-100, Agilent Technologies, Santa Clara, CA, USA, with supplements of 1 mM sodium pyruvate, 2 mM glutamine, 10 mM glucose and pH adjusted to 7.4). The glycolytic function was measured using the Seahorse XFp Glycolytic rate assay test kit (cat#103346-100, Agilent Technologies, Santa Clara, CA, USA). Briefly, the ECAR baseline readings were recorded using the Seahorse XFp analyzer (cat#431018, Agilent Technologies, Santa Clara, CA, USA), and the following injections were done with 4 µM Rot/AA and 50 mM 2-deoxyglucose (2-DG) respectively. PER, glycoPER, basal glycolysis, basal proton efflux rate, and compensatory glycolysis were calculated using the manufacturer’s formula.

### 2.9. Fuel Choice/Mito Fuel Flex Assay

For analysis of mitochondrial fuel oxidation pathways, the Seahorse XFp Mito Fuel Flex test (cat#103270-100, Agilent Technologies, Santa Clara, CA, USA) was performed according to the manufacturer’s instructions at the recommended drug concentrations, and the Seahorse XF Mito Fuel Report Generator was utilized to produce results.

### 2.10. Metabolite Measurement In Vitro

WERI-Rb1 cells of 1 × 10^6^ density per condition (post 72 h-transduction) were used for the measurement of metabolites like pyruvate (cat#MAK071, Sigma Aldrich, St. Louis, MO, USA), NAD+/NADH (cat#MAK037, Sigma Aldrich, St. Louis, MO, USA), ATP (cat#MAK190, Sigma Aldrich, St. Louis, MO, USA) and L-carnitine (cat#MAK063, Sigma Aldrich, St. Louis, MO, USA) according to the manufacturer’s protocol. For WERI-Rb1, cells (1 × 10^6^) cultured in one fuel source (glucose or glutamine) were used, followed by quantification of metabolites according to the manufacturer’s protocol.

### 2.11. Immunofluorescence/Mito Tracker Green

A total of 5 × 10^3^ WERI-Rb1 cells per transduced condition were seeded on 96-well plates (Eppendorf, Hamburg, Germany) pre-coated with poly-L-lysine. The cells were stained with 1:5000 dilution of Mito Tracker green (cat#M7514, Thermo Fischer Scientific, Waltham, MA, USA) in 10% RPMI media and Hoechst 33342 (cat#H1399, Invitrogen, Waltham, MA, USA) for 15 min. The images were captured using the ImageXpress High content confocal system (Molecular device, San Jose, CA, USA), and the mitochondrial intensity was calculated using MetaXpress software (Molecular device, San Jose, CA, USA).

For staining tumor tissues, 6 µm sections of Rb tumor (*n* = 5) and pediatric retina (*n* = 3) were deparaffinized in xylene (3 washes) and rehydrated in ethanol and water. They were then subjected to heat-induced epitope retrieval using citrate buffer (pH 6.0) for 20 min at 100 °C. The tissue sections were permeabilized using 0.1% Triton × 100 in 1 × PBS for 10 min at room temperature (RT) followed by 45 min of blocking in 3% BSA in 1 × PBS solution at RT. The tissue sections were further washed thrice in 1 × PBS for 5 min and were stained with 1:2000 dilution of Mito Tracker green (cat#M7514, Thermo Fischer Scientific, Waltham, MA, USA) in 1 × PBS for 20 min. The slides were mounted using Fluoroshield medium (cat#F6057, Sigma Aldrich, St. Louis, MO, USA) fortified with DAPI as a counterstain. The images were captured using the ImageXpress High content confocal system (Molecular device, San Jose, CA, USA), and the mitochondrial intensity was calculated using MetaXpress software (Molecular device, San Jose, CA, USA).

### 2.12. Statistical Analysis

Statistical analysis was performed using GraphPad Prism 8 (San Diego, CA, USA). Data are presented as mean ± s.d. unless indicated otherwise, and *p* < 0.05 was considered statistically significant. For all representative images, results were reproduced at least three times in independent experiments. For all quantitative data, the statistical test used is indicated in the legends. A statistical ‘decision tree’ is shown in Appendix A.

## 3. Results

### 3.1. RB1 Expression Reduces Mitochondrial Respiration in Retinoblastoma Cells

We have interrogated the consequence of Rb protein loss in patient tumors using transcriptomic profiling and identified significantly enriched glycolysis and reduced fatty acid metabolism pathways [17]. To model this altered metabolic profile, we tested the effects of RB1 complementation in different Rb null retinoblastoma cells. To elucidate how RB1 complementation affects mitochondrial function, we have used the Seahorse XFp analyzer to measure mitochondrial respiration in Rb null and Rb-complemented Y79, WERI-Rb1 and patient-derived, primary GL1-RB1 retinoblastoma cells. First, we ectopically expressed RB1 in WERI-Rb1 (Appendix A), Y79 (Appendix A) and GL1-RB1 (Appendix A). Further, we measured real-time OCR rates in Rb null and RB1-complemented WERI-Rb1 (Appendix A), Y79 (Appendix A) and GL1-RB1 (Appendix A) retinoblastoma cells. We found a significant reduction in mitochondrial respiration in RB1-overexpressed retinoblastoma cells compared to Rb null controls. Measuring mitochondrial respiration parameters revealed low basal mitochondrial respiration, maximal respiration and spare respiration in RB1-over-expressed WERI-Rb1 (Figure 1A–C), Y79 (Figure 1D–F) and GL1-RB1 (Figure 1G–I), further highlighting an altered mitochondrial function and energy profile in retinoblastoma cells with RB1 complementation.

### 3.2. RB1 and HK1 Complementation in Rb Null Cells Reveal Distinct Energy Profiles

In our previous study, we identified HK1 and E2F2 as two of the top differentially regulated genes in patient tumors when compared with matched pediatric, healthy retina controls [17]. Since HK1, which catalyzes the first step of glycolysis, represents a critical node of the cellular metabolic network and E2F2 is a less studied member of the E2F family, we investigated further the intracellular status of metabolism upon modulation of RB1, E2F2, and HK1. RB1 overexpression induced HK1 and reduced E2F2 protein levels in WERI-Rb1 cells. E2F2 overexpression and knockdown did not significantly induce HK1 protein, and similarly, HK1 modulation did not affect E2F2 protein levels (Figure 2A), indicating that they are independently regulated by Rb. Using energy phenotype tests, we measured the metabolic parameters in RB1- and HK1-modulated cells and found significant changes to their energy profile compared to controls. RB1-overexpressed cells showed a quiescent phenotype (Figure 2B) with significant incompetence in utilizing mitochondrial respiration under stressed conditions (OCR and ECAR of cells under induced energy demand) compared to energy-rich Rb null WERI-Rb1 cells (Figure 2E). Similarly, HK1-overexpressed cells with or without RB1 overexpression displayed a glycolytic phenotype with reduced utilization of mitochondrial respiration under an energy demand (Figure 2C,E). Conversely, HK1 knockdown cells showed an energy-rich phenotype with high competence in utilizing mitochondrial respiration rather than glycolysis under stressed conditions, which are restricted by RB1 complementation (Figure 2D,E). Collectively, we show that rescue of RB1 and HK1 in Rb null cells alters the metabolic profile towards an energy-restrictive glycolytic phenotype and away from mitochondrial respiration.

### 3.3. RB1 and HK1 Augmentation Reduce Mitochondrial Mass in Rb Null Cells

Since mitochondrial dynamics and, consequently, their mass affects mitochondrial respiration in tumors [22], we evaluated the active mitochondrial status in *RB1-* and *HK1*-complemented WERI-Rb1 cells using Mito-tracker dye. We observed lesser mitochondrial mass in *RB1-* and *HK1*-complemented cells compared to controls, while *HK1* knockdown showed a higher mitochondrial mass (Figure 3A,B). *HK1* overexpression led to a reduction in active mitochondria, explaining observations in previous reports [6,17]. Since cellular energy production through OXPHOS is dependent on mitochondrial status [23], we show that overexpression of *RB1* and *HK1* reduces mitochondrial mass in Rb null cells. We further validated the mitochondrial status in Rb tumor tissues (*n* = 5) and healthy, pediatric control retina (*n* = 3) using the Mito-tracker dye. We found a significantly higher overall mitochondrial mass in Rb tumor tissues compared to the photoreceptor layers of the pediatric retina (*p* = 0.04), where the mitochondria appear restricted to specific cellular layers (Figure 3C,D). Thus, higher mitochondrial mass in tumor tissues corroborates with elevated TCA and fatty-acid metabolism in Rb tumors, as reported previously by us and others.

### 3.4. HK1 and RB1 Expression Induce a Metabolic Switch from Mitochondrial Respiration to Glycolysis

The consequences of *HK1* and *RB1* expression modulation on energy production and mitochondrial function were validated using Seahorse XFp mito-stress assays (Figure 4A). Basal (Appendix A) and maximal (Appendix A) mitochondrial respiration was significantly suppressed upon ectopic expression of *RB1* and *HK1* either alone or in combination. Conversely, HK1 knockdown increased respiration, which was suppressed by *RB1* (Appendix A). Spare respiratory capacity was reduced significantly in *RB1-* and *HK1*-expressing cells (Figure 4B), indicating curtailed OXPHOS dependence. The spare respiratory capacity was higher in *HK1* knockdown cells, indicating that lack of HK1 in patient tumors possibly enhances their reliance on OXPHOS (Figure 4A,B). Consequently, both basal (Figure 4C,D) and compensatory (Figure 4E) glycolytic proton efflux rates (glycoPER) in cells expressing *RB1* and *HK1* were significantly elevated, indicating their metabolic shift towards glycolysis. However, the glycolytic capacity of *HK1*-ablated cells was significantly lower compared to control (Figure 4E), indicating that tumor cells have low dependence on glycolysis. *E2F2* overexpression and *E2F2* knockdown in these cells did not show any significant change in mitochondrial respiration parameters (Appendix A). This suggests that in the absence of *RB1* and low *HK1*, the Rb tumor cells acquire an ability to replenish the TCA cycle, even when glucose-derived pyruvate levels are low (Appendix A).

To analyze this aspect further, we measured key metabolites involved in glycolysis, the TCA cycle, and fatty acid metabolism in *RB1-, HK1-,* and *E2F2*-modulated cells. Pyruvate levels were significantly higher in *HK1*-overexpressing cells (similar to *RB1*) compared to *HK1*-ablated and control cells (Figure 4F). To evaluate the function of the mitochondrial TCA cycle-driven ETC (electron transport chain) function, we measured NAD+/NADH levels. *RB1* and *HK1* overexpression resulted in a significant reduction in NAD+/NADH levels compared to controls and *HK1*-ablated cells (Figure 4G). Further, L-carnitine, a key metabolite generated during β-oxidation of long-chain fatty acids [24] that produces higher ATP than glycolysis, was higher in controls and *HK1*-ablated cells compared to *RB1-* and *HK1-*overexpressing cells (Figure 4H). Consequently, ATP levels were significantly reduced in *RB1-* and *HK1*-expressing cells compared to controls. Conversely, ablation of *HK1* increased ATP output (Figure 4I). The NAD+/NADH ratios and L-carnitine levels indicate that in the absence of *RB1* and *HK1*, tumor cells can flexibly utilize alternate fuels such as glutamine or fatty acids to feed OXPHOS, fulfilling their bioenergetic demands. In all these experiments, *E2F2* modulation did not alter pyruvate, ATP, the NAD+/NADH ratio, or L-carnitine levels compared to control (Appendix A), demonstrating the specificity of the HK1 function.

### 3.5. RB1 Expression Restricts the Usage of Alternate Fuels in Rb Null Cells

The reliance of Rb null cells on alternate fuel sources like glutamine or fatty acids was evident from earlier experiments, and this ability can provide a ready supply of carbon substrates to the cancer cells for mitochondrial TCA anaplerosis and lipid generation [25]. Since the tumor transcriptome findings also revealed high expression of glutamine pathway genes [17], we tested the ability of WERI-Rb1, Y79, and GL1-RB1 cells to use alternate fuels in the presence or absence of *RB1* using the Seahorse mito-fuel flex assay. In the presence of *RB1*, cellular glucose dependency and flexibility were both significantly higher (Figure 5A and Appendix A). However, cells without *RB1* had significantly higher flexibility to use glutamine (Figure 5B and Appendix A) or fatty acids (Figure 5C and Appendix A), suggesting that Rb tumors use these alternate fuels for their growth.

To validate these findings, we measured key metabolite levels in Rb null and Rb-complemented WERI-Rb1 cells fed with glucose and glutamine. WERI-Rb1 and Y79 cells lacking *RB1* produced significantly lower pyruvate when fed with glutamine (*p* < 0.001) rather than glucose (Figure 5D and Appendix A) when compared to Rb-complemented cells. This corroborates with the results from patient vitreous humor (Appendix A), where pyruvate levels were lower. However, the total ATP (Figure 5G) and NAD+/NADH levels (Figure 5E and Appendix A) were higher in glutamine-fed conditions compared to glucose in WERI-Rb1 and Y79 cells, with a concurrent increase in L-carnitine (Figure 5F and Appendix A). This validates that Rb null cells rely on non-glucose fuel sources to replenish the mitochondrial TCA cycle and increase ATP production. Thus, in the absence of a functional Rb or reduced HK1, these tumor cells produce higher amounts of ATP through increased mitochondrial respiration fed by multiple fuel sources such as glutamine and fatty acids (Figure 5H).

## 4. Discussion

The present study identifies *RB1* and *HK1* as previously unrecognized regulators of metabolic adaptation in retinoblastoma cells. In studies, we have identified low expression of the glycolytic HK1 protein and pyruvate in Rb tumors, and we further revealed that ectopic expression of *RB1* induces HK1 in Rb null retinoblastoma cells [17]. We report that ectopic expression of *RB1* reduced mitochondrial respiration and spare respiration capacity of WERI-Rb1, Y79, and patient-derived GL1-RB1 retinoblastoma cells, highlighting that mitochondrial function is curtailed by the wild-type Rb protein. We show that Rb proteins diminish mitochondrial functions by inducing HK1, a rate-limiting glycolytic enzyme that phosphorylates glucose and can alter mitochondrial functions and metabolic pathways [26,27]. HK1 levels are significantly low in retinoblastoma tumors and *RB1*-null retinoblastoma cells [17], which is distinct from other types of solid tumors. High expression of HK1 is observed in various tumors [28,29], possibly driving higher glycolysis in those cases to fulfill the tumor cell’s energy demands. Contrary to these findings, we report that *RB1* and *HK1* complementation in Rb null cells drives them to adopt a glycolytic phenotype and restricts the high energy-producing mitochondrial respiration. We show that *RB1* and *HK1* complemented cells have reduced mitochondrial mass, which are characteristics of altered mitochondrial biogenesis [30], further highlighting the ability of these genes to impede OXPHOS and mediate a metabolic switch to glycolysis. Notably, loss of functional Rb in breast cancers enhanced mitochondrial proteins and OXPHOS with a subsequent increase in mitochondrial function [10], which agrees with our data, where Rb tumors possess high mitochondrial mass, in contrast to the healthy retina. In the retina, photoreceptor precursor cells are thought to be the cells of origin for Rb tumors [31]. Photoreceptors primarily rely on HK1 and aerobic glycolysis for rapid energy production [32,33], consuming a major portion of the glucose to produce lactate [34]. However, the whole retinal tissue depends on both glycolysis and mitochondrial OXPHOS for its functional and structural requirements to provide vision, which includes restricting the use of fatty acids potentially for the production of photoreceptor outer segment disks [34]. Since Rb tumor cells lack photoreceptor functions (as shown by significantly reduced expression of such genes [17]) and acquire higher proliferative potential, these cells reprogram their metabolism to meet the biosynthesis and energy production needs.

Retinoblastoma patient tumors have reduced pyruvate levels but higher levels of fatty acids. Fatty acids are known to enhance cancer proliferation by providing intermediates that are essential for maintaining cell membrane structure and function, energy storage, and signal transduction [16]. Their synthesis starts with the generation of mitochondrial acetyl CoA derived from citrate via pyruvate. However, in the absence of glycolytic pyruvate, the anaplerotic flux of carbon in the TCA cycle is maintained by glutamine for the generation of acetyl CoA [35]. We show that the lack of HK1 allows the Rb-null cancer cells to expand their flexibility of fuel choices, using glutamine and fatty acids for TCA anaplerosis. Such enhanced utilization of OXPHOS results in greater mitochondrial activity and higher ATP generation in tumor cells lacking *RB1* and *HK1*. Rescuing *RB1* or *HK1* in retinoblastoma cells shifted the cellular metabolic profile to glycolysis dependence, consequently reducing mitochondrial ATP production. Concurrent reduction in fatty acid pathway metabolite L-carnitine indicates that HK1-overexpressing cells no longer use fatty acid as a key energy source.

The restrictive metabolic profile directed by *RB1* and *HK1* in Rb cells can be exploited for pharmacological targeting in clinical settings. Although our investigations in the current study were limited to in vitro models, our findings on metabolic phenotypes of Rb null cells agree with the data obtained from primary patient samples and raise hopes for targeting Rb tumors. In conclusion, our work reveals the metabolic phenotypes and nutrient sensing in Rb null cells, illustrating *RB1-* and *HK1*-mediated metabolic flexibility as a potential therapeutic target.

## 5. Conclusions

In conclusion, we demonstrate Hexokinase 1 (HK1) to be significantly reduced in Rb tumors and *RB1* null retinoblastoma cells, wherein their loss mediates a critical metabolic adaptation towards enhanced mitochondrial respiration. Conversely, the data show a significant reduction in mitochondrial function and mass in Rb- and HK1-complemented retinoblastoma cells characterized by a metabolic switch from OXPHOS to glycolysis and reduced fuel utilization flexibility. Thus, our study provides a better understanding of the metabolic adaptations of retinoblastoma cells that can be useful for developing cancer metabolism-targeted therapeutics.

## Figures and Tables

**Figure 1 cells-11-03182-f001:**
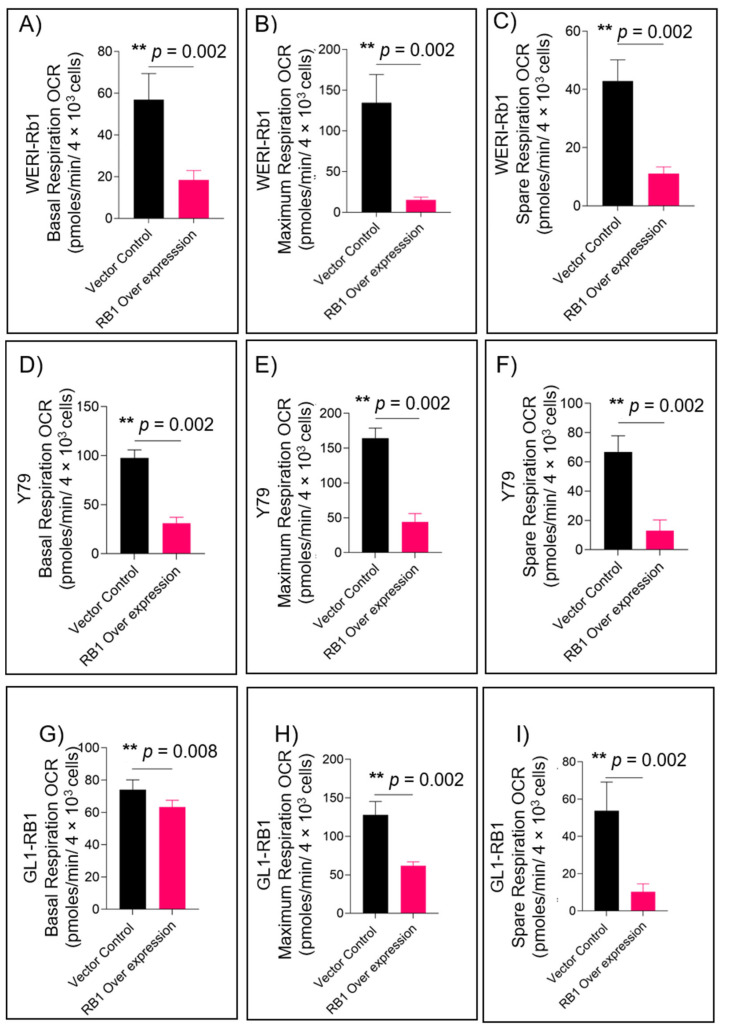
RB1 expression reduces mitochondrial respiration in retinoblastoma cells. Seahorse XFp mito-stress test assay showing (**A**) basal respiration, (**B**) maximum respiration, and (**C**) spare respiration in WERI-Rb1 cells; (**D**) basal respiration, (**E**) maximum respiration, and (**F**) spare respiration in Y79 cells; and (**G**) basal respiration, (**H**) maximum respiration, and (**I**) spare respiration in GL1-RB1 cells. Values represent the mean ± s.d. of three independent experiments. Two-tailed Mann–Whitney test was used for statistical analysis. ** *p* < 0.01.

**Figure 2 cells-11-03182-f002:**
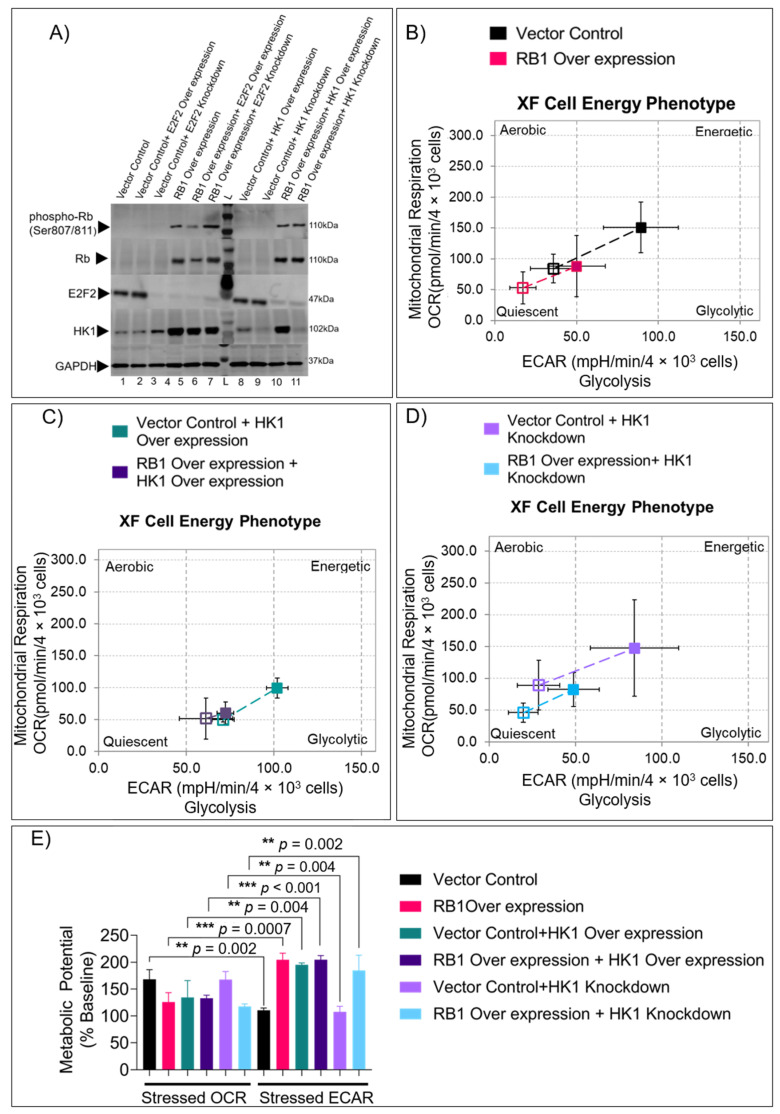
*RB1* and *HK1* complementation in Rb null cells reveals distinct energy profiles. (**A**) Western blot showing the protein expression of phospho-Rb, Rb, E2F2, and HK1 in *RB1* null and *RB1*-overexpressed (OE) WERI-Rb1 cells modulated with E2F2 OE and HK1 OE and knockdown constructs. Energy phenotype profile of (**B**) vector control and *RB1*-overexpressed WERI-Rb1 cells, (**C**) *HK1* OE cells vs. *RB1* OE+ *HK1* OE WERI-Rb1 cells, and (**D**) *HK1* knockdown vs. *RB1* OE+ *HK1* KD cells. (**E**) Metabolic potential of modulated cells under stressed phenotype. Open squares represent baseline activity, and closed squares represent stress-induced activity. Values represent three independent experiments mean with data points ± s.d. Kruskal–Wallis with Dunn’s multiple comparisons test (for >2 groups) were used for statistical analysis. ** *p* < 0.01, *** *p* < 0.001. OE = overexpression; KD = knockdown.

**Figure 3 cells-11-03182-f003:**
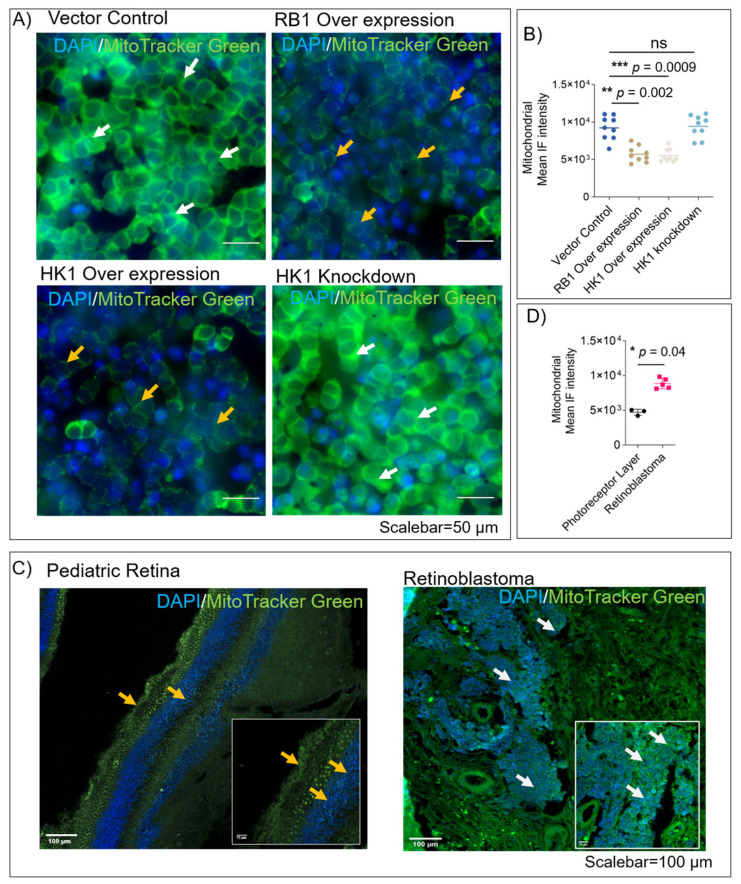
RB1 and HK1 augmentation reduce mitochondrial mass in Rb null cells. (**A**) Mitochondrial density in cells transduced with indicated vectors modulating *RB1* and *HK1* as indicated; post 96-h transduction, the cells were stained with Mito-tracker green and Hoechst 33342 for 10 min. White arrows indicate high mitochondrial mass, while yellow arrows indicate low mitochondrial mass. Scale bar = 50 µm. (**B**) Mean mitochondrial IF intensity was calculated per condition using HCS and MetaXpress software. Immunofluorescence showing mitochondrial density in (**C**) photoreceptor layers of the pediatric retina (*n* = 3) and B) Rb tumor tissue (*n* = 5) using Mito-tracker green dye. Mean IF intensity of mitochondria, (**D**) pediatric retina and Rb tumors. White arrows indicate high mitochondrial mass, while yellow arrows indicate low mitochondrial mass. Scale bar = 100 µm. Values represent three independent experiments mean with data points ± s.d. Two-tailed Mann–Whitney test (for 2 groups) or Kruskal–Wallis with Dunn’s multiple comparisons test (for >2 groups) were used for statistical analysis. * *p* < 0.05, ** *p* < 0.01, *** *p* < 0.001.

**Figure 4 cells-11-03182-f004:**
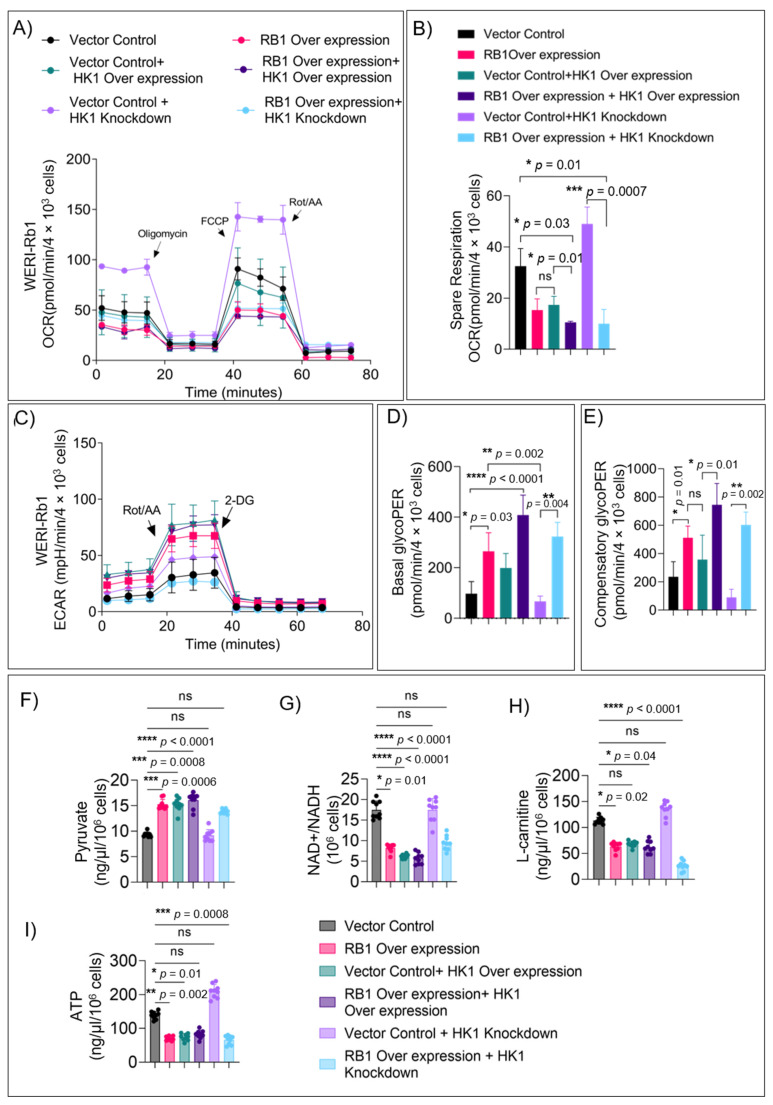
HK1 and RB1 expression induce a metabolic switch from mitochondrial respiration to glycolysis. (**A**) WERI-Rb1 cells were transduced with indicated vectors modulating RB1 and HK1 as indicated, and oxygen consumption rate (OCR) was measured using Seahorse XFp Mito-stress assay to assess mitochondrial function in WERI-Rb1 cells. (**B**) Spare respiration. (**C**) Seahorse XFp glycolytic rate assay to measure glycolysis in cells transduced with different vectors as indicated. (**D**) Basal glycolysis. (**E**) Compensatory glycolysis. In cells transduced with indicated vectors, measurement of key metabolites were assessed: (**F**) pyruvate levels indicating glycolytic flux (**G**) NAD+/NADH ratios to estimate mitochondrial TCA functions, (**H**) L-carnitine levels to evaluate dependency on fatty acid oxidation, (**I**) total ATP levels to estimate energy produced. Values represent three independent experiments mean with data points ± s.d. Two-tailed Mann–Whitney test (for 2 groups) or Kruskal–Wallis with Dunn’s multiple comparisons test (for >2 groups) were used for statistical analysis. * *p* < 0.05, ** *p* < 0.01, *** *p* < 0.001, **** *p* < 0.0001.

**Figure 5 cells-11-03182-f005:**
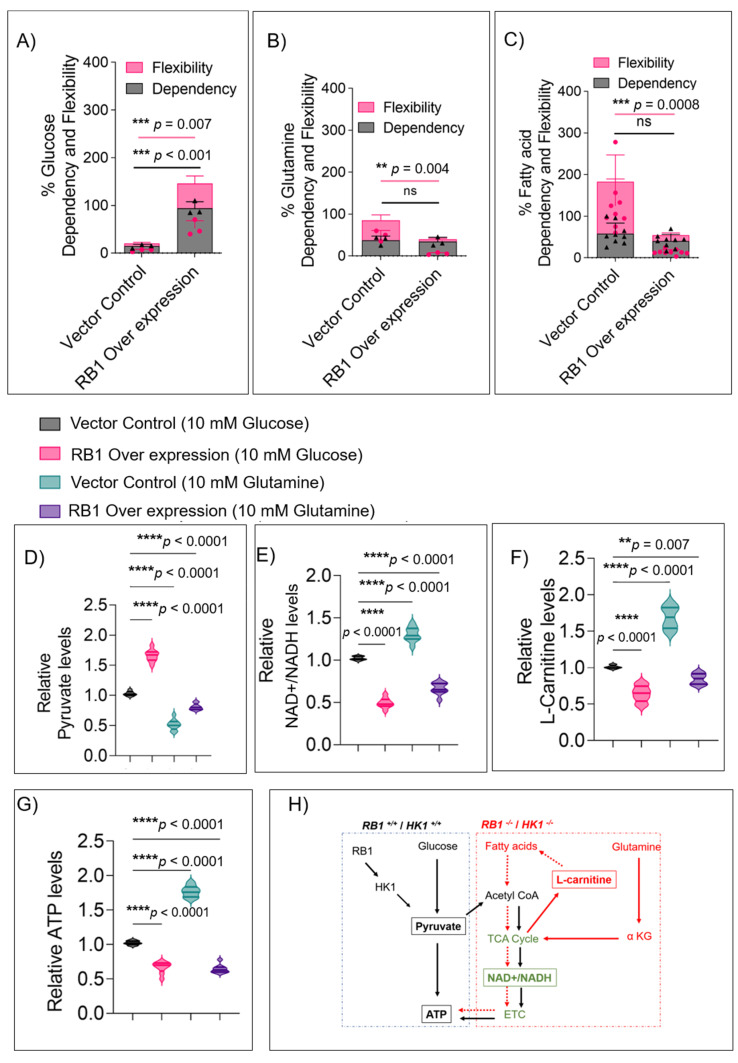
*RB1* expression restricts the usage of alternate fuels. Seahorse XFp Mito-fuel flex assays to measure fuel dependency and flexibility in cells transduced with different vectors as indicated. (**A**) Oxidation of glucose as fuel in control and *RB1*-complemented WERI-Rb1 cells (*n* = 3). (**B**) Oxidation of glutamine as fuel (*n* = 3). (**C**) Oxidation of fatty acids as fuel (*n* = 3). Validation of fuel oxidation profile in cells cultured in glucose and glutamine alone conditions and transduced with indicated vectors. Measurement of key metabolites was assessed as follows: (**D**) pyruvate levels indicating glycolytic flux, (**E**) NAD+/NADH ratios to estimate mitochondrial TCA functions, (**F**) L-carnitine levels to evaluate dependency on fatty acid oxidation. (**G**) Total ATP levels to estimate energy produced. (**H**) Schematic representation of metabolites in each metabolic pathway under *RB1* null and *RB1-*overexpressed cells. Values represent the mean ± s.d. of three independent experiments. Two-tailed Mann–Whitney test (for 2 groups) or Kruskal–Wallis with Dunn’s multiple comparisons test (for >2 groups) were used for statistical analysis. ** *p* < 0.01, *** *p* < 0.001, **** *p* < 0.0001.

**Table 1 cells-11-03182-t001:** Clinical and histopathological details of samples used for validations.

ID	Sex	Age at Presentation	Laterality	Clinical Risk	IIRC Group	AJCC Staging
P1	F	23 months	Bilateral	Advanced	Group E	cT3b
P2	F	24 months	Unilateral	Advanced	Group E	cT3b
P3	M	36 months	Bilateral	Advanced	Group E	cT3b
P4	F	33 months	Unilateral	Non-advanced Group D	cT2b	33 months
P5	F	14 months	Bilateral	Non-advanced Group D	cT2b	14 months
Control 1	F	3 months	NA	Cardiac arrest (no ocular complications)		
Control 2	F	2 months	NA	Multiple organ dysfunction (no ocular complications)		
Control 3	M	6 months	NA	No ocular complication		

**Table 2 cells-11-03182-t002:** Details of qPCR primers used in the study.

Gene	Forward Primer	Reverse Primer	Tm(F/R)
RB1	TTTGTAACGGGAGTCGGGA	CAGCGAGCTGTGGAGGAG	54.67/55.89
β-Actin	TCCCTGGAGAAGAGCTACGA	AGGAAGGAAGGCTGGAAGAG	56.9/55.2

## Data Availability

Data will be available on request.

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
