# Peer review of "Lack of Retinoblastoma Protein Shifts Tumor Metabolism from Glycolysis to OXPHOS and Allows the Use of Alternate Fuels"

_cells, 2022, doi:10.3390/cells11203182_

Round 1

Reviewer 1 Report

The manuscript titled “Lack of Retinoblastoma protein shifts tumor metabolism from glycolysis to OXPHOS and allows the use of alternate fuels” describes tumor suppressor retinoblastoma protein plays an important role in the tumor metabolism between glycolysis and OXPHOS energy generation. More details need to provide. The followings are some concerns and comments have been pointed out that the authors may want to consider.

1) Line 28: It should be “NAD+” instead of only “NAD”.

2) Line 35 Keywords: The keyword “ECAR” only appears one time in the main content, “Energetics” or “bioenergetics” or “energetic” appears five times in the main content. I’d suggest the authors consider some other suitable one if you don’t mind.

3) Line 56: The reference [18] published on 1982 is not a recent work of the authors’ group. Additionally, who is the co-author of both [18] and the current manuscript, please?

4) Line 68 Materials and Methods section: a) Please include cat# etc. information for all the reagents used in this study throughout the manuscript. b) Please include at least a brief description of each protocol to make your work relatively easier to repeat.

5) Line 77: Please be consistent with or without a space between “value” and “unit” throughout the manuscript.

6) Line 92: The “2” should be subscript for “CO2”. Check throughout the manuscript.

7) Line 99: Please incorrect input “xxC(t)”.

8) Line 101 Table 2: a) Please include directions. b) Please add “-“ following “β”.

9) Line 103 lentiviral plasmids and vectors section: Please include more details to make your work relatively easier to reproduce. For example, lentiviral concentration, and so on.

10) Line 113: The number “6” should be superscript. Check throughout the manuscript.

11) Line 132: Please be consistent with “h” or “hrs”.

12) Line 136: Please include “poly-L-lysine” concentration.

13) Line 154: Please extend “OCR” as it appears the first time.

14) Line 199: Please use italic p throughout the manuscript as it refers to a p-value.

15) Line 226: What’s the meaning of “group>2” for the “two-tailed Mann-Whitney test”? There are only two groups for each panel image.

16) Line 241: Please specify “stressed conditions”.

17) Line 251 Figure 2: In Figure 1A western blot lanes 6 and 7, it seems E2F2 overexpression significantly suppressed p-Rb(Ser807/811). Are there any analysis and description of this?

18) Line 364 Figure 5: Please provide a higher resolution Figure 5H.

19) Did the authors perform one-way ANOVA? Is there any difference between one-way ANOVA with Tukey posttest of multiple comparisons and Kruskal–Wallis with Dunn’s multiple comparisons test?

20) Are there any specific metabolic inhibitors that have been used in the related study to further confirm the results?

21) Where is the Rb knockdown/silencing data? I can’t find them. The related experiments are important to support your hypothesis. 

Reviewer 2 Report

Rb is a major tumor suppressor protein commonly lost in several cancers. Rb1 loss has also been associated with treatment induced plasticity in cancers like small cell lung cancer and neuroendocrine prostate cancer which further highlight the importance of studying and understanding the effect of Rb1 loss in cancer cells. In this manuscript, authors analyzed the role of Rb1 on mitochondrial metabolism in retinoblastoma cancer using patient tissue samples and cell lines. Although the study is clean and showed very clearly that Rb1 loss has been associated with metabolic shift from glycolysis to OXPHOS in Rb; the major criticism is that this finding is not novel. There are numerous studies reported the same in several cancer types (Reviewed in Trends in cancer. PMID: 29120753).

Other major concerns are-

1.       One of the key findings in this manuscript is that Rb1 loss provide more flexibility to Rb cancer cells in terms of utilization of energy source as shown in Section 3.5 and Fig 5 which is interesting. However, authors performed the experiments in only one cell line WERI-Rb1. One cell line is not enough to draw these conclusions and authors should perform similar experiments in at least two other cell lines (preferably Y79 and GL1-RB1) for rigor.

2.       Mitotracker green dye binds with free thiol groups in mitochondria which allow its retention in mitochondria even after the depolarization or loss of membrane potential. Therefore, this dye is not a good system to analyze mitochondrial activity as mentioned by authors in section 3.3 and Fig 3. Mitotracker Green dye is good to analyze mitochondrial mass as mentioned by authors but cannot be used to analyze mitochondrial activity. Authors should consider this while interpretating this data.

3.       Further, Rb1 overexpression decreased mitochondrial mass as shown in Fig 3A and because the used dye cannot differentiate between active and inactive mitochondria, it will be obvious that mitochondrial OXPHOS activity will be down. This observed phenomenon may be primarily due to decreased mitochondrial mass following Rb1 overexpression.

4.       Authors should include the detailed methodology of Normal and Rb tumor tissue staining with mitotracker dyes in Section 2 (Material and methods).

5.       As shown in Fig 4A, HK1 ablation enhance OCR in Rb cells then why NAD/NADH ratio is same in HK1 ablated and vector control cells (Fig 4G). An explanation is needed.

6.       Page 7, Line 235. Authors mentioned that E2F2 knockdown did not induce HK1 protein. However, the results shown in Fig 2A suggest HK1 upregulation following E2F2 knockdown (Lane 1 vs Lane 3).

Round 2

Reviewer 1 Report

Thank you for the update. Please consider the following concerns. The statistical information is very important. Lots of comments that the authors did not make any modifications, please check the supplementary file as well. PLEASE CHECK THROUGHOUT THE MANUSCRIPT INSTEAD OF ONLY THE POINTS BELOW.

1) Line 219: Please italic p as it refers to a p-value. Check throughout the manuscript including supplementary.

2) Line 310: Please indicate the meaning of “2<group”. Check throughout the manuscript. This is important; there are a lot.

3) Figure S2: Please include statistical information.

4) Figure S3: It should be “NAD+” instead of only “NAD”.

5) Supplementary file: Please be consistent with or without a space between “value” and “unit” throughout the manuscript.

6) Please make it clearer; what’s statistical method has been used for the two groups’ comparison?

Reviewer 2 Report

Authors revised the manuscript satisfactorily and manuscript may be accepted for publication.
